# ScalableMap: Scalable Map Learning for Online Long-Range Vectorized HD Map Construction

**Jingyi Yu**
School of Geodesy and Geomatics
Wuhan University
jingyiyu@whu.edu.cn

**Zizhao Zhang**
Electronic Information School
Wuhan University
3zair1997@gmail.com

**Shengfu Xia**
School of Geodesy and Geomatics
Wuhan University
xiashengfu@whu.edu.cn

**Jizhang Sang**
School of Geodesy and Geomatics
Wuhan University
sangjzh@whu.edu.cn

**Abstract:** We propose a novel end-to-end pipeline for online long-range vectorized high-definition (HD) map construction using on-board camera sensors. The vectorized representation of HD maps, employing polylines and polygons to represent map elements, is widely used by downstream tasks. However, previous schemes designed with reference to dynamic object detection overlook the structural constraints within linear map elements, resulting in performance degradation in long-range scenarios. In this paper, we exploit the properties of map elements to improve the performance of map construction. We extract more accurate bird's eye view (BEV) features guided by their linear structure, and then propose a hierarchical sparse map representation to further leverage the scalability of vectorized map elements, and design a progressive decoding mechanism and a supervision strategy based on this representation. Our approach, ScalableMap, demonstrates superior performance on the nuScenes dataset, especially in long-range scenarios, surpassing previous state-of-the-art model by 6.5 mAP while achieving 18.3 FPS. Code is available at https://github.com/jingy1yu/ScalableMap.

**Keywords:** Map Construction, Multi-view Perception, Long-range Perception

## 1 Introduction

To ensure the safety of autonomous vehicles on the road, downstream tasks such as trajectory prediction and motion planning typically rely on high-definition (HD) maps as prior information [1, 2], which provide centimeter-level location information for map elements. However, the production of such HD maps is generally carried out offline, involving complex processes known for their high labor and economic costs, making it difficult to construct maps that cover a wide area [3, 4].

Recent researches aim to construct local maps in real-time using on-board sensors. More studies show the superiority of schemes based on bird's-eye view (BEV) representation for unifying data from various sensors. Early attempts [5, 6, 7, 8, 9] regard map construction as a semantic segmentation task, using convolutional neural networks (CNN) to obtain occupancy grid map. However, these schemes can only generate rasterized maps, which lack instance and structure information about map elements and are therefore difficult to apply directly to downstream tasks [10].

HDMapNet [11] uses time-consuming heuristic post-processing algorithms to generate vectorized maps. More recent approaches [12, 13] focus on constructing end-to-end networks similar to dynamic object detection schemes, treating map elements as ordered sets of vertices. However, the properties of map elements are different from those of dynamic objects. Map elements are typically

7th Conference on Robot Learning (CoRL 2023), Atlanta, USA.

linear and often parallel to axes [14], which makes it difficult to define bounding boxes. Moreover, in dense vehicle scenarios, the limited visibility of vertices with map elements in the image space hinders accurate map shape inference solely based on heatmaps. While a recent approach [15] proposes hierarchical query embeddings to better describe the arbitrary shape of an element by modeling each vertex as a query, it requires dense points to ensure the shape of elements and need to predict a large number of vertices simultaneously without structural guidance. This poses challenges to the convergence speed and performance, particularly in long-range scenarios. Therefore, there is still a need for an approach that can effectively capture the structural constraints within map elements to achieve high accuracy in long-range HD map construction tasks.

In this paper, we aim to exploit the structural properties of vectorized map elements to address the challenges of accurately detecting map elements at longer ranges. First, we extract position-aware BEV features and instance-aware BEV features via two branches respectively and fuse them under the guidance of linear structure to get hybrid BEV features. Next, we propose a hierarchical sparse map representation (HSMR) to abstract map elements in a sparse but accurate manner. Integrating this representation with cascaded decoding layers proposed by DETR [16], we design a progressive decoder to enhance the constraints of structured information by exploiting the scalability of vectorized map elements and a progressive supervision strategy to improve the accuracy of inference. Our scheme, ScalableMap, dynamically increases the sampling density of map to get inference results at various scales, allowing us to obtain more accurate map information faster.

**Contributions.** Our contributions are summarized as follows: (i) We propose ScalableMap, a first end-to-end long-range vectorized map construction pipeline. We exploit the structural properties of map elements to extract more accurate BEV features, propose a HSMR based on the scalable vectorized elements, and design a progressive decoder and supervision strategy accordingly. All of these result in superior long-range map perception. (ii) We evaluate the performance of ScalableMap on the nuScenes dataset [17] through extensive experiments. Our proposed approach achieves state-of-the-art results in long-range HD map learning, surpassing existing multimodal methods by 6.5 mAP while achieving 18.3 FPS.

## 2   Related work

**Lane Detection**   The lane detection task has been a popular research topic for many years. Early approaches to these tasks [18, 5] usually rely on segmentation schemes that require complex post-processing to obtain the final result. In order to obtain structured information, some schemes [19, 20] aim to find a unified representation of curves, while others [21, 22, 23, 24] utilize anchor-based schemes to abstract map elements with open shapes. Compared with the above solutions, our thinking is closer to HRAN [4], which directly outputs structured polylines. However, it relies on a recurrent network that is known to be computationally inefficient. Our ScalableMap is capable of handling real map elements with complex geometric structures, while the previously mentioned methods can only handle a single type or regular shape.

**Boundary Extraction**   Boundary extraction aims to predict polygon boundaries for object instances on images. Polygon-RNN [25, 26] adopts recurrent structure to trace each boundary sequentially, which is not suitable for scenarios with real-time requirements. Some works [22, 27, 28] achieve good results in boundary extraction, but they are generally designed for polygons in image space and are not suitable for map construction tasks. The closest to our proposed scheme is BoundaryFormer [29], which uses queries to predict vertices of polygons to obtain vectorized polygon boundaries. However, the differentiable loss it defines for closed-shape elements in image space is not suitable for map element that is dominated by open-shape linear elements, as they have less concentrated features compared to dynamic objects.

**Vectorized HD Map Construction**   Recent work tries to infer vectorized HD maps directly from on-board sensor data. HDMapNet [11] generates vectorized maps using a time-consuming heuristic

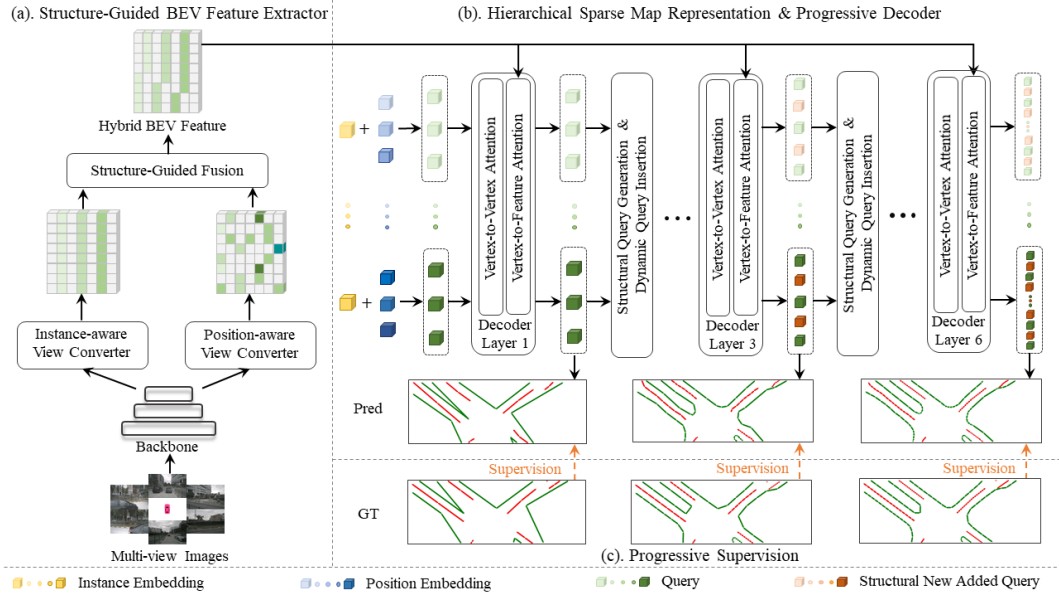

Figure 1: Overview of ScalableMap. (a) Structure-guided hybrid BEV feature extractor. (b) Hierarchical sparse map representation & Progressive decoder. (c) Progressive supervision.

post-processing method, while VectorMapNet [12] proposes a two-stage framework with an end-to-end pipeline using a slow auto-regressive decoder to recurrently predict vertices. InstaGraM [13] proposes a graph modeling approach based on vertex and edge heatmaps to reason about instance-vertex relations, which may be difficult to infer some vertices of a map element appeared in multiple views. Given the challenge of dealing with arbitrary shapes and varying numbers of vertices in elements, MapTR [15] tackles this by employing a fixed number of interpolations to obtain a uniform representation. But MapTR's hierarchical query design primarily focuses on the structural association of elements during the initialization phase, resulting in slow convergence and deteriorating performance as the perception range increases. Only SuperFusion [30] is a relevant work for long-range vectorized HD map construction, which also uses post-processing to obtain vectorized results. Our model is the first end-to-end scheme that utilizes the structural properties of map elements throughout the entire process to construct long-range vectorized maps.

## 3 Methodology

### 3.1 Overview

Given a set of surround-view images $\{I_1, ..., I_k\}$ captured from $k$ on-board cameras, the goal of ScalableMap is to predict $M$ local map elements $\{L^{(j)}; j = 1, ..., M\}$ within a certain range in real-time, including lane dividers, road boundaries, and pedestrian crossings. Each map element is represented by a sparse set of ordered vertices, which can be described as $L^{(j)} = \{(x_0, y_0, z_0), ..., (x_{m_j}, y_{m_j}, z_{m_j})\}$, where $m_j$ is the number of vertices of element $L^{(j)}$ and $(x, y, z)$ are the coordinates of each vertex in a unified vehicle coordinate system.

The architecture of ScalableMap is illustrated in Figure 1. We build the model with three components to construct long-range vectorized HD maps: (1) *structure-guided hybrid BEV feature extractor*: transforming camera sensor data into BEV features with structure-guided fusion (Section 3.2) ; (2) *progressive decoder*: layer by layer map element decoding based on the proposed HSMR (Section 3.3) ; (3) *progressive supervision*: bipartite matching and training for HSMR (Section 3.4).

## 3.2 BEV Feature Extractor

The ill-posed nature of 2D-3D transformation is exacerbated by the elongated and linear characteristics of map elements, leading to feature misalignment and discontinuity. To obtain hybrid BEV features, we utilize one branch for extracting position-aware BEV features and another branch for extracting instance-aware BEV features. These branches are then fused together, guided by the structural properties of map elements.

**Perspective View Converter.** We start by extracting image features through ResNet. Method proposed by BEVFormer [31] is adopted to obtain position-aware BEV features $F_{bev}^p$, which utilizes deformable attention [32] to enable spatial interaction between BEV queries and corresponding image features based on a predefined 3D grid and calibration parameters. Additionally, we use several multi-layer perception (MLP) layers to obtain instance-aware BEV features $F_{bev}^i$ since they are effective at preserving continuous features in image space [33]. $k$ image features are individually converted to their respective top-views using $k$ MLPs. To further improve feature continuity across views, we use a linear layer to transform top-view features into a unified BEV feature.

**Structure-Guided Feature Fusion.** To enhance the robustness of features for accurate map construction, we propose a mutual correction strategy that leverages information from two distinct features: $F_{bev}^p$ with relatively precise positional data for select map vertices, and $F_{bev}^i$ encompassing comprehensive shape information for map elements. By directly summing these features, we produce the updated $F_{bev}^{i'}$. Additionally, we introduce a segmentation head to $F_{bev}^{i'}$, guiding it to focus on the drivable area to learn the transformation scale. Subsequently, $F_{bev}^p$ is concatenated with the refined $F_{bev}^{i'}$, and their fusion is executed through a convolutional layer. This fusion process corrects misalignments in $F_{bev}^p$, producing an hybrid BEV feature with enhanced richness and accuracy.

## 3.3 Progressive Decoder

The varied shapes of vectorized map elements present challenges for conventional abstraction schemes like bounding box-based and anchor-based approaches. To address this, we introduce a HSMR as the core idea of our approach. HSMR provides a sparse and unified representation that accurately describes the actual shape of elements while supporting fast inference. Building upon this, we design a progressive decoder inspired by the DETR paradigm. Moreover, we incorporate a module that generates structural queries first and then dynamically inserts queries, acting as a vital bridge to connect maps of different densities.

**Hierarchical Sparse Map Representation.** Polyline representations of map elements are typically obtained by sampling points where the curvature exceeds a threshold, thus resulting in varying numbers of vertices for each element. We define the number of vertices forming each element as the map density to ensure a consistent representation. Based on this density, we employ uniform point sampling for elements with an excessive number of vertices, while for elements with fewer vertices than the desired density, we perform point subsampling based on distances between the original vertices. This approach allows us to obtain representations of the same element at arbitrary densities. By combining the iterative optimization idea of DETR paradigm with the dynamically adjustable density of the vectorized map, we hierarchically utilize a low-density map as an abstract representation of the high-density map. The low-density map captures map element shapes adequately while being sufficiently sparse. A visual depiction of HSMR and its performance is provided in Figure 4.

**Decoder Layers.** We define query $q_{n,m}$ responsible for the $m$-th vertex of the $n$-th element. Leveraging the hierarchical sparse representation of map elements, a small number of queries are initially generated to capture the approximate shape of each map element. Each query is formed by adding an instance embedding $q_n^{ins}$ and a position embedding $q_{n,m}^{pos}$. Our progressive map element decoder is composed of multiple decoder layers, each containing two types of attention mechanisms. These attention mechanisms facilitate information exchange among vertices, and enable interaction between

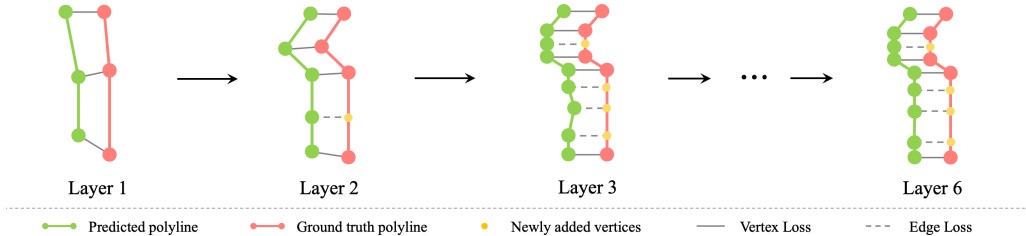

Figure 2: Visualization of progressive polyline loss.

each vertex and its corresponding BEV feature. The exchange between vertices is implemented using multi-head self-attention [34], while the other is implemented using deformable attention [32].

**Structural Query Generation and Dynamic Query Insertion.** To connect layers that handle different densities, we exploit the positional constraints among adjacent vertices within the same element to augment the map density. We introduce new queries by taking the mean value of two adjacent queries that share an edge, and dynamically insert new queries between these two queries. Rather than employing traditional methods that initialize a large number of queries simultaneously and update them iteratively, we adopt a strategy of initializing each element with only a limited number of queries and gradually increasing the map density layer by layer. This enables the module to focus on the original sparse instance features and leverage the structural characteristics of vectorized map elements, ensuring robust long-range perceptual capabilities.

### 3.4  Progressive Supervision

During training, we infers $N$ map elements $\{\hat{L}^i\}_{i=1}^N$ in each layer, and $N$ is set to be larger than the typical number of elements in a scene. Assume there are $M$ targets $\{L^i\}_{i=1}^M$, which is padded with $\emptyset$ to form a set of size $N$. Following [16, 35], bipartite matching is employed to search for a permutation $\sigma \in \Sigma$ with the lowest cost. $\Sigma$ includes the equivalent permutation for each element, as multiple vertex orders can represent the actual shape of an element in map construction task:

$$\sigma^* = argmin_{\sigma \in \Sigma} \sum_{i=1}^{N} [-\mathbb{1}_{\{c_i \neq \emptyset\}} \hat{p}_{\sigma(i)}(c_i) + \mathbb{1}_{\{c_i \neq \emptyset\}} \mathcal{L}_{match}(\hat{L}_{\sigma(i)}, L_i)] \tag{1}$$

where $\hat{p}_{\sigma(i)}(c_i)$ is the probability of class $c_i$ for the prediction with index $\sigma(i)$ and $\mathcal{L}_{match}$ is a pair-wise polyline matching cost between prediction $\hat{L}_{\sigma(i)}$ and ground truth $L_i$. We use Hungarian algorithm [36] to find the optimal assignment $\sigma^*$. We employ focal loss to supervise the element category and drivable area, and additional loss terms are incorporated in the following loss function:

$$\mathcal{L}_{polyline} = \lambda_v \mathcal{L}_{vertex} + \mathcal{L}_{edge} \tag{2}$$

**Vertex Loss.** Considering HSMR involves subsampling process, we differentiate the supervision between original vertices and newly added vertices. A visual representation of the supervision mechanism for the progressive polyline loss is shown in Figure 2. For each of predicted original vertex $\hat{v}_{\sigma*(i),j}$ assigned to vertex $v_{i,j}$, we employ L1 distance to ensure prediction accuracy. With $N_v^l$ standing for the number of original vertices in layer $l$, vertex loss of each element is formulated as:

$$\mathcal{L}_{vertex} = \sum_{i=1}^{N} \mathbb{1}_{\{c_i \neq \emptyset\}} \sum_{j=0}^{N_v^l - 1} \|\hat{v}_{\sigma*(i),j} - v_{i,j}\|_1 \tag{3}$$

**Edge Loss.** We use edge loss to supervise edge shape, which includes distance from newly added vertices $\{\hat{v}_{\sigma(i),j,k}\}_{k=1}^{N_v^j - 1}$ corresponding to the current edge $e_{i,j}$, edge slope, and angle formed by adjacent edges. The distance component is supervised with L1 loss, while the slope and angle

Table 1: Results on nuScenes validation dataset. C denotes camera only and C+L denotes camera-LiDAR fusion. Range represents the perceived range along the Y-axis. FPS of ScalableMap is measured on a single RTX 3090 GPU, with batch size 1 and GPU warm-up. The metrics of MapTR* are obtained by retraining the model while modifying only the perception range, following the official code and ensuring consistency with the claimed specifications. Metric values marked with † represents the AP value under a threshold of 1.0. Since SuperFusion only provides this metric, we conduct the same benchmark test for a fair comparison.

| Method | Modality | Range | $AP_{ped}$ | $AP_{divider}$ | $AP_{boundary}$ | mAP | FPS |
|---|---|---|---|---|---|---|---|
| HDMapNet | C | [−30.0, 30.0] | 14.4 | 21.7 | 33.0 | 23.0 | 0.8 |
| HDMapNet | C+L | [−30.0, 30.0] | 16.3 | 29.6 | 46.7 | 31.0 | 0.5 |
| VectorMapNet | C | [−30.0, 30.0] | 36.1 | 47.3 | 39.3 | 40.9 | 2.9 |
| VectorMapNet | C+L | [−30.0, 30.0] | 37.6 | 50.5 | 47.5 | 45.2 | - |
| InstaGraM | C | [−30.0, 30.0] | 47.2 | 33.8 | 44.0 | 41.7 | 17.6 |
| MapTR | C | [−30.0, 30.0] | 56.2 | 59.8 | 60.1 | 58.7 | 11.2 |
| **ScalableMap** | C | [−30.0, 30.0] | **57.3** | **60.9** | **63.8** | **60.6** | **18.3** |
| MapTR* | C | [−60.0, 60.0] | 35.6 | 46.0 | 35.7 | 39.1 | 11.2 |
| **ScalableMap** | C | [−60.0, 60.0] | **44.8** | **49.0** | **43.1** | **45.6** | **18.3** |
| SuperFusion | C+L | [0.0, 60.0] | 22.3† | 30.3† | **53.4†** | 35.3† | - |
| **ScalableMap** | C | [0.0, 60.0] | **51.0†** | **55.1†** | 48.4† | **51.5†** | **18.3** |

components are supervised with cosine similarity. The edge loss of each element is formulated as:

$$\mathcal{L}_e = \sum_{i=1}^{N} \mathbb{1}_{\{c_i \neq \emptyset\}} \{ \sum_{j=0}^{N_v^l-1} [\lambda_p \sum_{k=0}^{N_{v,j}-1} d(\hat{v}_{\sigma^*(i),j,k}, e_{i,j}) + \lambda_s c(\hat{e}_{\sigma^*(i),j}, e_{i,j})] + \lambda_a \sum_{j=0}^{N_v-2} c(\hat{a}_{\sigma^*(i),j}, a_{i,j}) \}$$

(4)

where $\hat{e}_{\sigma^*(i),j}$ is the edge formed by two adjacent vertices, $\hat{a}_{\sigma^*(i),j}$ is the angle formed by two adjacent edges, $d(\hat{v}, e)$ denotes the distance from vertex $v$ to edge $e$, and $c(\hat{a}, a)$ denotes the cosine similarity between two edges. $N_{v,j}$ is the number of added vertices corresponding to edge $e_{\sigma^*(i),j}$.

## 4 Experiments

### 4.1 Experimental Settings

**Dataset and Metrics.** We evaluate ScalableMap on the nuScenes dataset, which consists of 1000 scenes. Each scene has a duration of approximately 20 seconds. The dataset provides a 360-degree field of view around an ego-car, captured by six cameras. Following previous works [12, 13, 15], we use the average precision(AP) metric to evaluate the performance, and chamfer distance to determine which positive matches the ground truth. We calculate the AP for the three categories, and for each category, AP is computed under several thresholds $\{0.5, 1.0, 1.5\}$.

**Implementation Details.** We train ScalableMap for 110 epochs on RTX 3090 GPUs with batch size 32. The perception range for regular range test is $[-15.0m, 15.0m]$ along the $X$-axis and $[-30.0m, 30.0m]$ along the $Y$-axis, and the perception range for long-range test is expanded to $[-60.0m, 60.0m]$ along the $Y$-axis. To unify the representation of vertices, we use the Ramer–Douglas–Peucker algorithm [37] to simplify the original polyline with a threshold of $0.05m$ before subsampling. For training, we set the loss scales $\lambda_{cls}$ as 2.0, $\lambda_v$, $\lambda_p$ as 5.0 and $\lambda_s$, $\lambda_a$ as $5e^{-3}$ respectively. The progressive decoder is composed of six decoder layers.

### 4.2 Results.

**Comparison with Baselines.** We evaluate the performance of ScalableMap by comparing it with that of state-of-the-art methods on nuScenes validation test. As shown in Table 1, under camera modality, ScalableMap performs slightly better than MapTR, achieving 1.9 higher mAP and faster

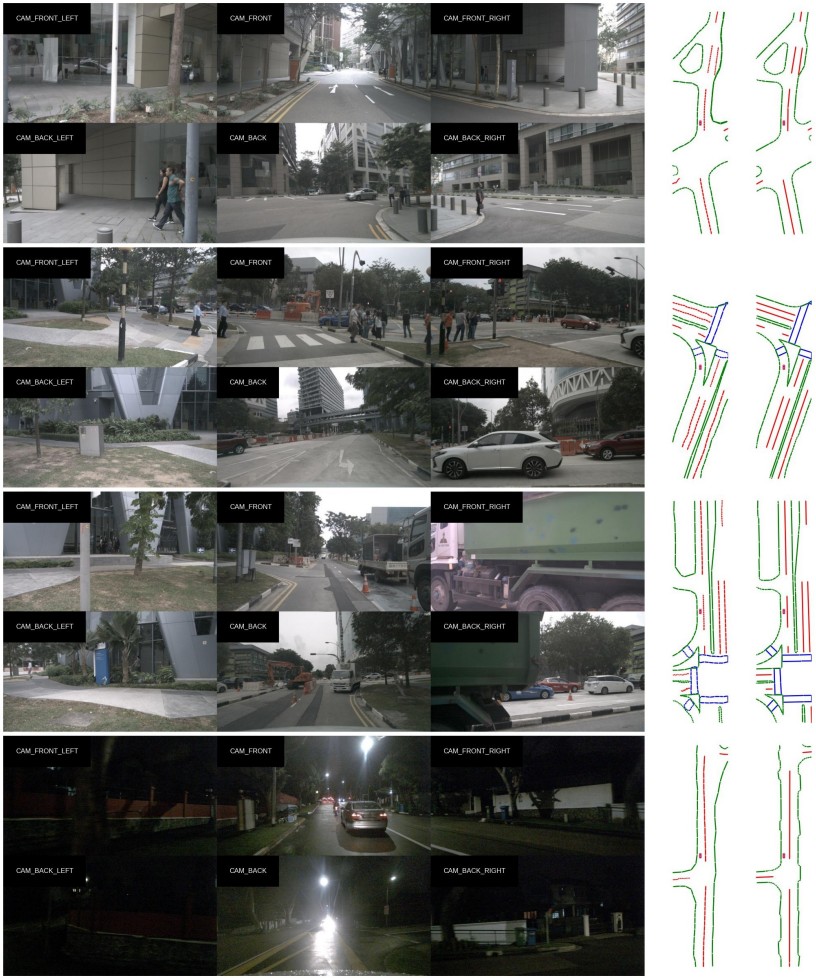

Figure 3: Visualization of qualitative results of ScalableMap in challenging scenes from nuScenes validation dataset. The left column is the surround views, the middle column is the inference results of the ScalableMap, the right column is corresponding ground truth. Green lines indicate boundaries, red lines indicate lane dividers, and blue lines indicate pedestrian crossings.

inference speed within the conventional perception range of $[-30.0m, 30.0m]$ along the Y-axis. When the same models are directly applied to $[-60.0m, 60.0m]$ scenario, ScalableMap achieves 45.6 mAP and 18.3 FPS, while MapTR's corresponding values are 39.1 and 11.2. It is noted that SuperFusion is the only method which publishes experiment results in this range. However, it is a fusion model of lidar and single-view camera. The mAP achieved by our approach is higher than that of SuperFusion by 16.2 under the same benchmark, demonstrating the superior performance even in a multi-view camera modality with near real-time inference speed. The results demonstrate that our scheme effectively meets the real-time requirements of online map construction tasks, delivering superior accuracy in both conventional perception range tests and long-range tests.

**Qualitative Results Visualization.** The visualization of qualitative results of ScalableMap on nuScenes validation dataset in long-range test is shown in Figure 3. More visualization results of challenging scenarios are presented in Appendix B for more visualization results of challenging scenarios. Our model still performs well even in curved roads, intersections, congested roads, and night scenes. We further visualize three out of six decoder layers of MapTR* and ScalableMap in Figure 4. Our strategy demonstrates a faster ability to focus on the instance features, while the progressive iteration yields more precise shapes of elements.

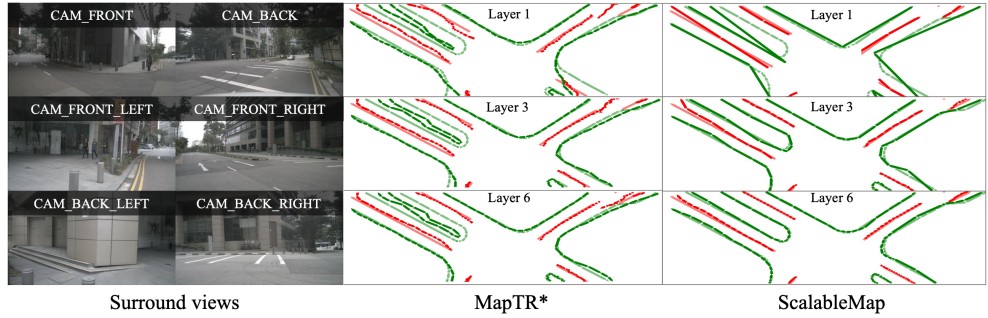

| Surround views | MapTR* | ScalableMap |

Figure 4: Visualization of prediction from three decoder layers of MapTR* and ScalableMap. The perception range along the Y-axis is $[-60.0m, 60.0m]$. The light-colored lines on the image represent ground truth, while the dark-colored lines represent the inference results.

## 4.3 Ablation Studies

We conduct ablation experiments on nuScenes validation set to verify the effectiveness of the components of the proposed method and different design. Settings of all experiments are kept the same as mentioned before. Additional ablation experiments are provided in Appendix A.

**Ablation of Proposed Components.** Table 2 presents experimental results showcasing the impact of our proposed components. HSMR demonstrates effective performance in long-range perception with sparse representation. SQG&DQI enhances structural information within map elements, while the SGFF module significantly enhances performance.

Table 2: Ablations about modules.

| HSMR | SQG&DQI | SGFF | mAP |
|:---:|:---:|:---:|:---:|
| | | | 40.1 |
| ✓ | | | 39.7 |
| ✓ | ✓ | | 42.6 |
| ✓ | ✓ | ✓ | **45.6** |

**Ablation of Number of Vertices.** Ablations of the effect of number of vertices forming each element on long-range perception in each decoder layer are presented in Table 3. The experimental results show that, based on our proposed HSMR, the model performance is quite stable with the number of vertices. We trade-off accuracy and speed to select the appropriate parameters.

Table 3: Ablations about vertex number.

| Number of Vertices | mAP | FPS |
|:---:|:---:|:---:|
| 2/3/5/9/9/9 | 43.6 | **19.2** |
| 3/5/9/17/17/17 | **45.6** | 18.3 |
| 4/7/13/25/25/25 | 44.2 | 17.6 |

## 5 Discussion

We propose ScalableMap, an innovative pipeline for constructing long-range vectorized HD maps. We exploit the inherent structure of map elements to extract accurate BEV features, propose the concept of HSMR based on the scalable vectorized maps, and design progressive decoder and supervision strategy accordingly to ensure fast convergence. Through these designs, our method effectively captures information over long distances. Experiment results on nuScenes dataset demonstrate compelling performance, particularly in long-range scenarios, thus affirming its real-time applicability and effectiveness in real-world environments.

**Limitations.** Our method relies solely on real-time camera sensor data, thus its performance depends on the visibility of the scenarios, which may be limited in situations like traffic congestion or extreme weather conditions. Additionally, accurate camera calibration parameters are assumed, which can pose a constraint in practical deployment. Future research can focus on reducing the reliance on calibration parameters by developing calibration-free approaches or incorporating online calibration methods. Exploring the integration of positional constraints among map elements or leveraging global coarse maps as prior knowledge may further enhance the robustness and accuracy.

**Acknowledgments**

We appreciate the reviewers for their comments and feedback.

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

# A  Ablation Study

## A.1  The Way of Feature Fusion.

Given that SGFF employs a mutual correction strategy, we conduct ablation experiments to validate the efficacy of this feature fusion approach. Specifically, we consider two scenarios: first, without correcting the position-aware features, where the two features are directly combined for fusion; and second, without correcting the instance-aware features, where the two features are directly fed into the convolutional layer for fusion. The result of these experiments robustly underscore the effectiveness of SGFF.

Table 4: Ablation studies on feature fusion approaches.

| Fusion Method | Ped Crossing | Divider | Boundary | mAP |
|---|---|---|---|---|
| w/o position-aware feature correction | 39.9 | 43.7 | 38.6 | 40.7 |
| w/o instance-aware feature correction | 42.3 | 46.0 | 39.0 | 42.4 |
| SGFF | 44.8 | 49.0 | 43.1 | 45.6 |

## A.2  Effectiveness of Edge Loss

Edge loss in ScalableMap comprises three crucial elements: the loss associated with newly introduced vertices concerning their corresponding edges, the loss concerning edge slopes, and the loss encompassing the angle formed by three consecutive vertices. The first element holds particular significance as it directly influences the shape regression of the map element, while the latter two elements exert their influence indirectly on the map element's shape. We conduct ablation experiments to underscore their efficacy in the context of map construction tasks.

Table 5: Ablation studies on edge loss.

| Loss Item | Ped Crossing | Divider | Boundary | mAP |
|---|---|---|---|---|
| w/o edge loss | 43.9 | 47.6 | 42.7 | 44.7 |
| with edge loss | 44.8 | 49.0 | 43.1 | 45.6 |

## A.3  Influence of Vertex Count on Convergence Speed

Figure 5 illustrates the convergence curves of our model in three ablation experiments conducted under different vertex configurations for long-range tests. Excessive vertices can hinder convergence, whereas insufficient counts can compromise the accuracy of shape representation. By carefully fine-tuning the number of vertices through ablation experiments, we strive to strike a balance. This fundamental approach aligns closely with the nuanced perspective presented in our paper, further strengthening the robustness of our findings.

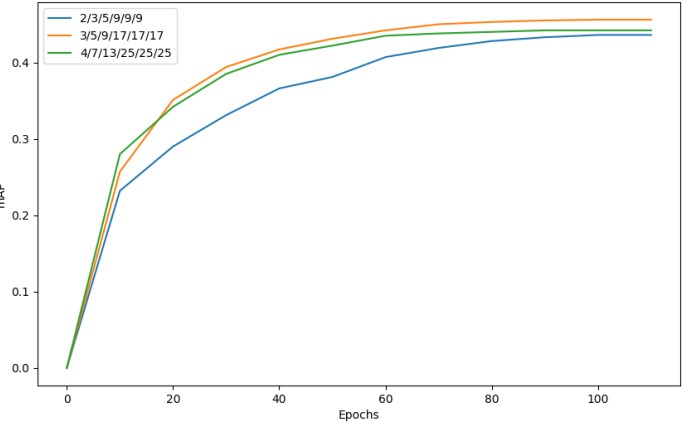

Figure 5: Visualization of convergence curves.

# B    Qualitative visualization

We present visual results of ScalableMap operating in adverse weather conditions and dealing with occlusion scenarios on nuScenes validation set as shown in Figure 6, Figure 7 and Figure 8.

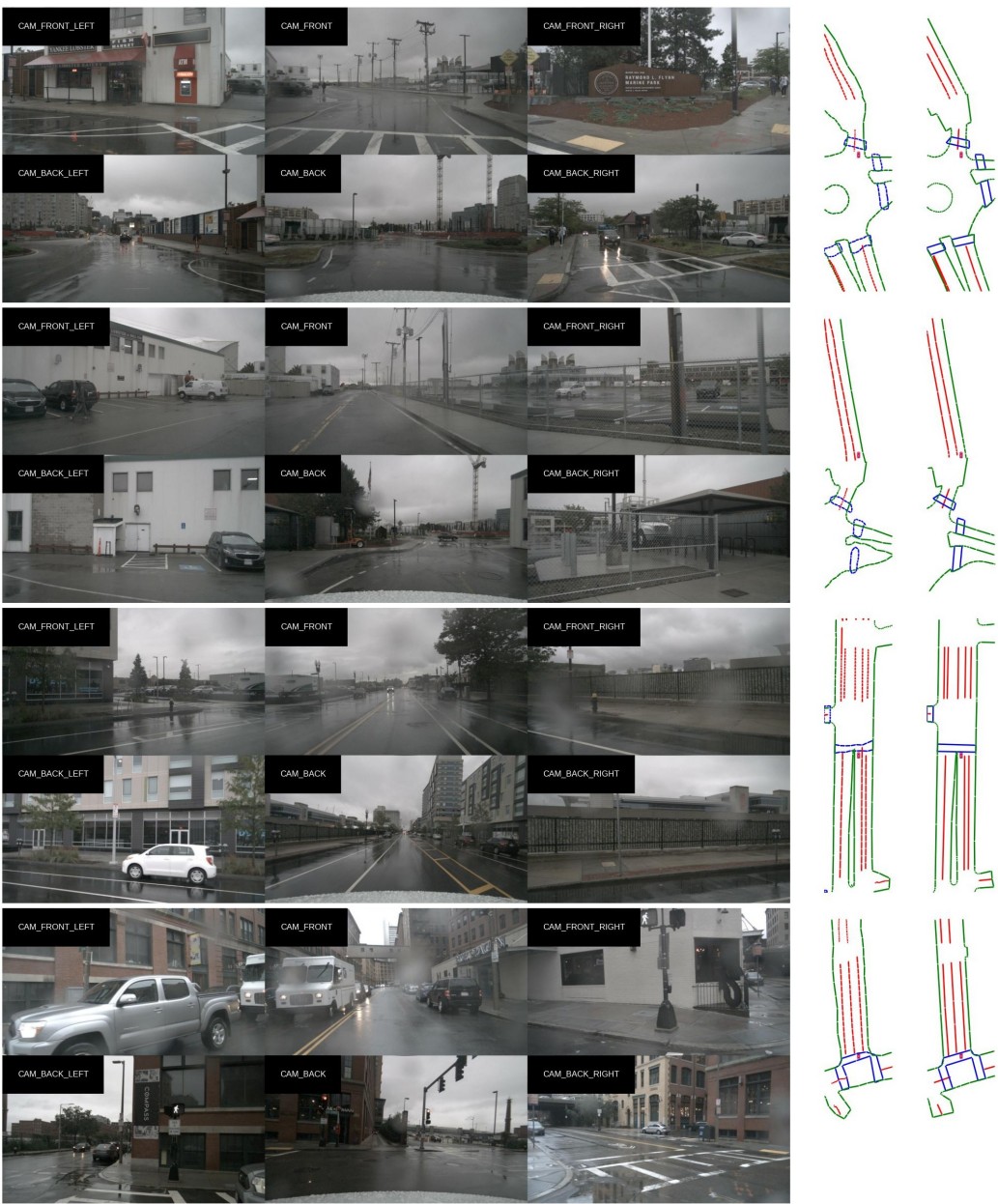

Figure 6: Visualization of qualitative results of ScalableMap in rainy scenes from nuScenes validation dataset. The left column is the surround views, the middle column is the inference results of the ScalableMap, the right column is corresponding ground truth. Green lines indicate boundaries, red lines indicate lane dividers, and blue lines indicate pedestrian crossings.

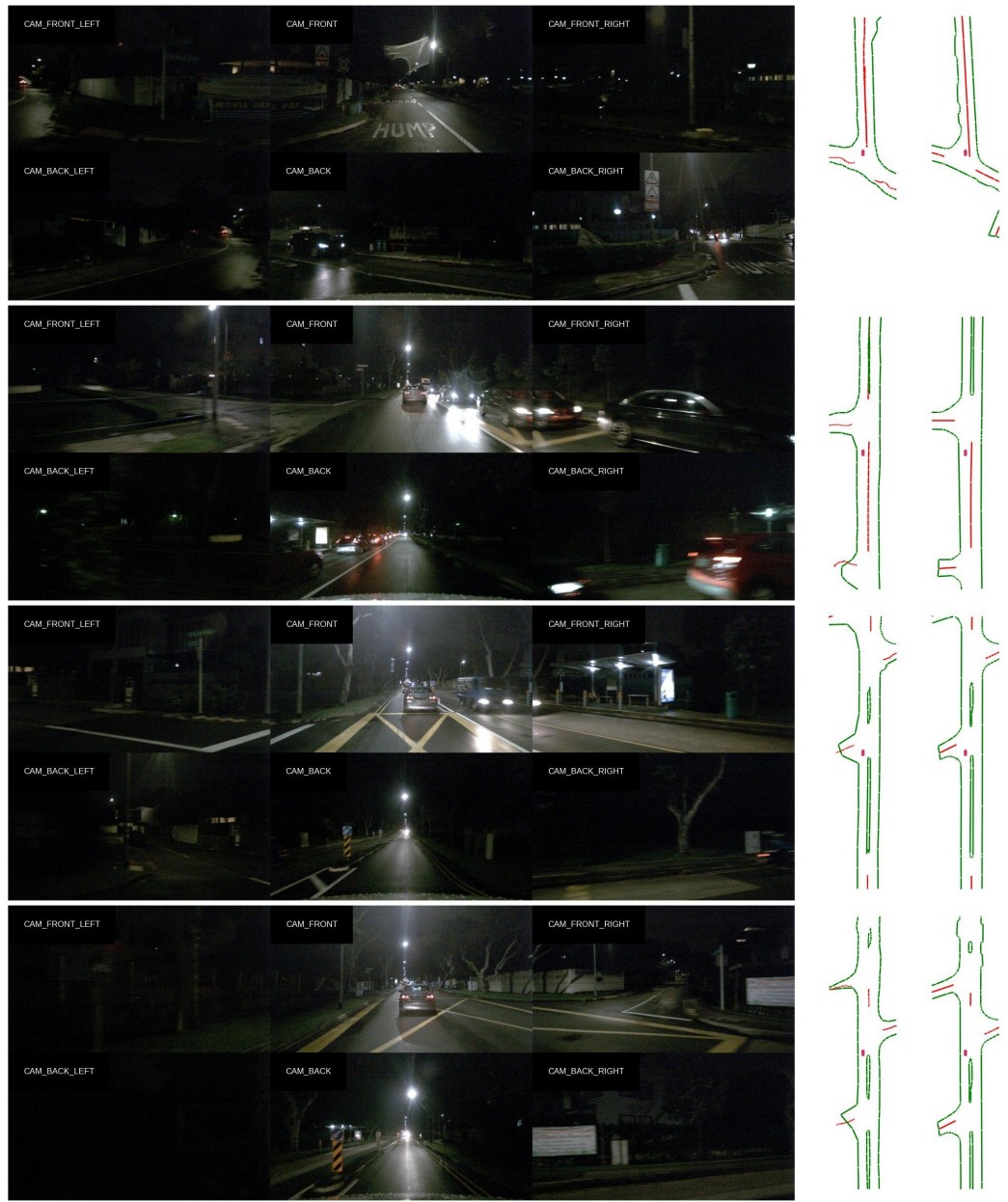

Figure 7: Visualization of qualitative results of ScalableMap in nightly scenes from nuScenes validation dataset. The left column is the surround views, the middle column is the inference results of the ScalableMap, the right column is corresponding ground truth. Green lines indicate boundaries, red lines indicate lane dividers, and blue lines indicate pedestrian crossings.

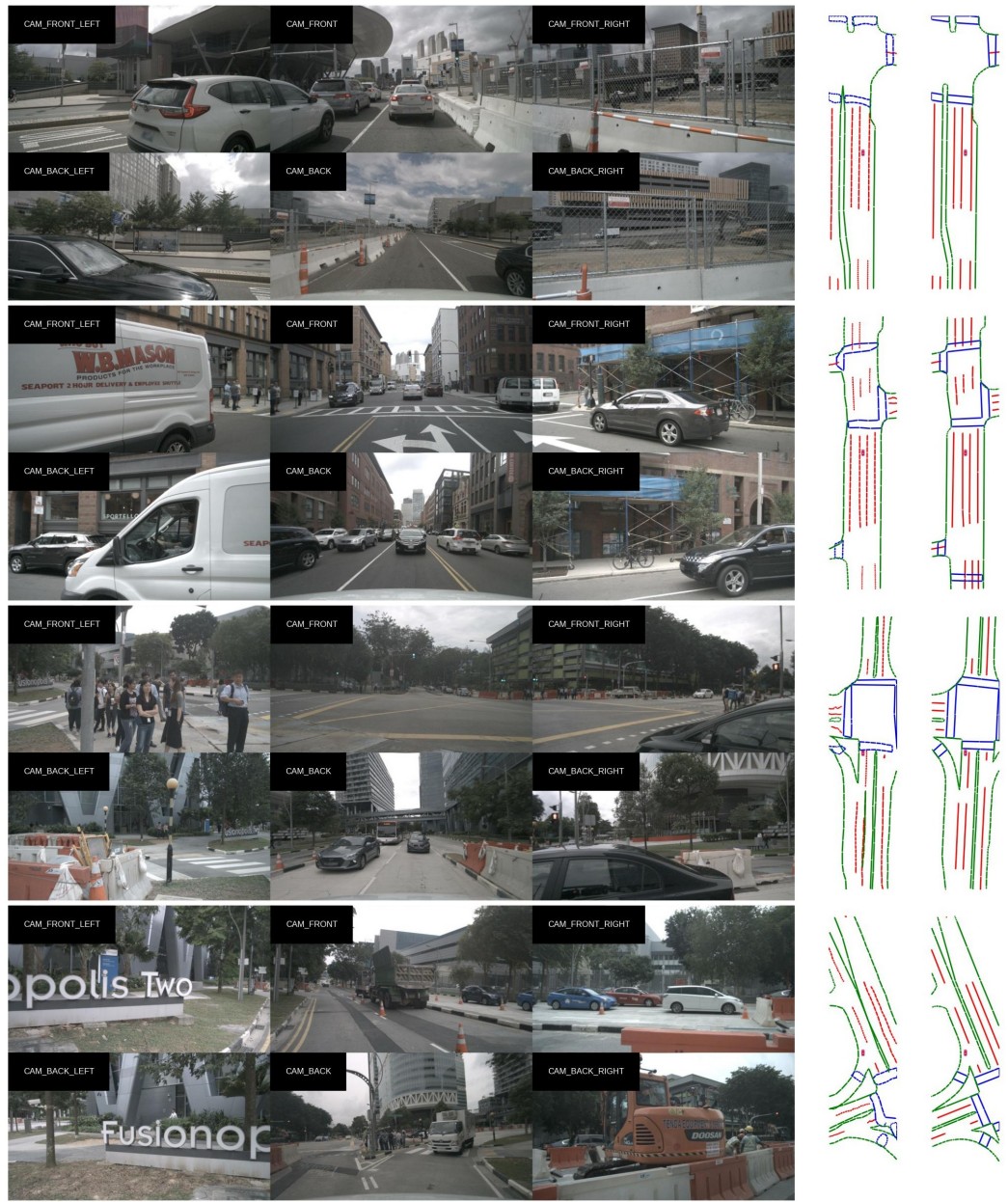

Figure 8: Visualization of qualitative results of ScalableMap in occlusion scenes from nuScenes validation dataset. The left column is the surround views, the middle column is the inference results of the ScalableMap, the right column is corresponding ground truth. Green lines indicate boundaries, red lines indicate lane dividers, and blue lines indicate pedestrian crossings.

