# OpenReview forum: "ScalableMap: Scalable Map Learning for Online Long-Range Vectorized HD Map Construction"
_robot-learning.org/CoRL/2023/Conference — CoRL 2023 Poster_

### Official Review · Reviewer_8yNs · 2023-07-17

**Confidence:** 5
**Originality:** Good
**Technical Quality:** Good
**Clarity Of Presentation:** Very Good
**Impact:** 3

**Recommendation:**

Weak Accept: I recommend accepting the paper, but will not argue for my recommendation if the majority of other reviewers have a different opinion.

**Review:**

1. This paper works on a vibrant new research topic of vectorized HD map construction and presents two method improvements, namely structure-aware BEV encoder and progressive decoder, to increase the performance upon previous SOTA methods.
2. HSMR is an interesting design for map elements that always come in different densities. I suspect it would be very useful for map learning. However the ablation study is missing in Table 2.
3. In the normal experiment setting of vectorized map construction, ScalableMap shows a relative minor improvement over MapTR (1.9 mAP).
4. Little rationality is provided about why ScalableMap is much better in long range. I am wondering whether other SOTA methods would have a much better result in long range (e.g., InstaGraM, VectorMapNet)
5. feature aggregation in SGFF is a bit arbitrary without sufficient ablation study.

**Quality Of The Limitations Section:**

Limitations are addressed clearly

**Questions For Rebuttal:**

My concerns are listed in the 'review' section. I would encourage authors to address my concerns.

**Robotics Focus:**

Highly relevant to robotics but no hardware experiments

**Summary Of Paper:**

The authors propose ScalableMap, a new methodology aimed at leveraging the structural properties of vectorized map elements for accurate long-range detection. This approach includes the extraction of position-aware and instance-aware bird's-eye view (BEV) features, which are fused using linear structure guidance to yield hybrid BEV features. A important part of this method is a hierarchical sparse map representation (HSMR), allowing for accurate but sparse abstraction of map elements. When combined with cascaded decoding layers (DETR), a progressive decoder is designed to enhance structured information constraints, using the scalability of vectorized map elements and a progressive supervision strategy to increase inference accuracy.

**Summary Of Recommendation:**

Overall, I think it's an interesting paper working on an important problem with two model improvements. Some of the design choices seem a bit arbitrary for me, thus I would encourage the author to add more experiments.

---

> ### Author Response · Authors · 2023-08-13
> **Response to Reviewer 8yNs**
>
> **Q3.** Little rationality is provided about why ScalableMap is much better in long range. I am wondering whether other SOTA methods would have a much better result in long range (e.g., InstaGraM, VectorMapNet).
>
> **A3.** We appreciate your insightful inquiry. Your concern regarding the rationale behind ScalableMap's superior performance in long-range scenarios is well-taken. Allow us to provide a comprehensive elucidation on this matter, addressing both the underpinning design philosophy and the comparison with other SOTA methods.
>
> ScalableMap's efficacy in long-range perception is grounded in its deliberate design philosophy, which prioritizes the incorporation of often overlooked structural properties within vectorized map elements. As the perceptual range extends, the significance of these structural associations becomes more pronounced. We visualize the results of previous work where many vertex predictions overlap with multiple elements, underscoring the importance of considering the element's structural context for holistic scene understanding, particularly in extended perceptual ranges. Our appraoch leverages multiple strategies to reinforce this aspect: the fusion of instance-aware and position-aware feature representations, the integration of hierarchical sparse map representation, and the implementation of structural query generation and dynamic query insertion strategies.
>
> In response to your inquiry about comparing ScalableMap with InstaGraM and VectorMapNet, we acknowledge the importance of rigorous comparisons. However, certain limitations constrain our ability to conduct exhaustive evaluations. The lack of publicly available source code for InstaGraM hampers comprehensive assessments against it. As for VectorMapNet, we are actively testing and intend to publish its results on our GitHub repository, accompanied by our official code. The performance of models tends to decrease with an increase in perceptual range. Our validation results demonstrate that ScalableMap's performance surpasses both VectorMapNet and InstaGraM across both normal and long-range scenarios. Thus, we believe that even when applied to long-range scenarios, ScalableMap's performance remains superior.
>
> Your feedback is invaluable, and we are grateful for the opportunity to clarify these aspects of our research.
>
> **Q4.** Feature aggregation in SGFF is a bit arbitrary without sufficient ablation study.
>
> **A4.** We greatly appreciate your feedback and would like to provide further clarification on the feature aggregation process within our SGFF approach. In our paper, we introduced two distinct types of BEV features obtained through separate viewpoint converters. These features offer unique insights into map element instances, capturing both their spatial distribution and approximate quantity. However, each feature also possesses inherent limitations. The instance-aware feature extractor, while informative about spatial distribution, lacks precise positional accuracy. Conversely, the position-aware feature extractor, due to inherent errors introduced during intrinsic and extrinsic projection processes, deviates from ground truth contour information and lacks global context.
>
> To obtain BEV features with stronger expressive ability without introducing more noise, we performed position correction operations on the two features respectively. We first use the position-aware BEV feature to correct the feature distribution on the instance-aware BEV feature and supervise the drivable area using a segmentation head, which will help inform the instance-aware feature about the true scale transformation. We then concatenate this feature with the original position-aware BEV feature to obtain a BEV feature that is more accurate in both local position and overall shape. The underlying principle of SGFF is grounded in leveraging the unique characteristics of map elements to enhance the overall expressiveness of BEV features.
>
> We have included a supplementary ablation experiment in our work, demonstrating the effectiveness of our feature fusion approach. This ablation experiment verifies the performance of feature fusion using one modality alone, and they are both degraded compared to SGFF, which validates the effectiveness of our modular design. `We have revised the module's description in the paper and supplemented corresponding ablation experiments in Appendix B.1.` We are committed to continuously refining our approach and appreciate your valuable input.
> | Fusion Method | Ped Crossing | Divider | Boundary | mAP |
> | :-------:|:--------:|:--------:|:--------:|:--------:|
> | w/o position-aware feature correction | 39.9    | 43.7     | 38.6     | 40.7 |
> | w/o instance-aware feature correction | 42.3    | 46.0     | 39.0     | 42.4 |
> | SGFF | 44.8    | 49.0     | 43.1     | 45.6 |
>
> We thank you again for your careful review. Please let us know if this addresses all of your questions and suggestions.

---

### Official Review · Reviewer_3QMy · 2023-07-19

**Confidence:** 4
**Originality:** Good
**Technical Quality:** Good
**Clarity Of Presentation:** Good
**Impact:** 3

**Recommendation:**

Weak Accept: I recommend accepting the paper, but will not argue for my recommendation if the majority of other reviewers have a different opinion.

**Review:**

Generally, I like the overall idea and premise of the paper of carrying the vectorized nature of the map elements throughout the pipeline. Also, I especially like the DETR approach of progressively decoding the output to higher vertex densities, where preceding stages provide rough alignment with subsequent refinement stages. I can easily imagine that this approach allows for good guidance and would help convergence.

I think the paper provides some good novelty towards end-to-end on-board mapping, and the results are definitely convincing. Unfortunately, the authors could not find the necessary space to explain their method in sufficient detail, and I keep wondering about the implementation details of many different pieces. I also checked the supplement (which contains the source code), but I would have greatly appreciated some more in-depth explanations in a separate appendix.

In the same vein, I think that the evaluation is generally weak. The comparison was conducted on a single dataset only, and the ablation provided only very little insight. I understand that the page limit is tight, but IMO the authors wasted too much space with Figure 1 (could be removed), and Figure 3 (could be shortened).

Also, Section 3.4 on Progressive Supervision, as well as the preceding section on the structural and dynamic queries, is not completely clear to me after multiple readings, and I suggest adding at least one figure that visualizes these various things.

Although the writing is overall clear, there are occasionally some missing words or slight grammatical mistakes. I would ask the authors to proof-read the Method section, and fix the cite issue on line 189.

**Quality Of The Limitations Section:**

Limitations are addressed clearly

**Questions For Rebuttal:**

Concerning the iterative/progressive refinement: Table 3 shows a sudden mAP drop in the last row. What would explain that? For example, are higher vertex densities in the first stages posing more difficulty for the losses to converge? Also, since the last three progression rounds have the same number of vertices per element, have you seen a real necessity to keep the 6 stages? Unfortunately, there is no ablation on this.

In lines 196-198 you mentioned that you simplify the curves to unify the vertex representation. Can you explain what unification means in this context (i.e. different densities for different levels)? And do you do this only for training, or also when comparing against the GT? Also, do you think that the 5cm threshold could introduce noticeable errors?

The polyline loss seems interesting enough, but I see no clear benefit of the edge loss without any ablation study. Are there pathological cases where the vertex loss would fail to optimize towards the optimal configuration, and only the edge loss would provide enough signal to resolve a tie? Or is the idea to compensate for induced errors in GT resampling?

**Robotics Focus:**

Highly relevant to robotics but no hardware experiments

**Summary Of Paper:**

The paper addresses the problem of camera-based onboard mapping by means of detecting vectorized road map elements. The authors propose a method that is based on three separate stages, where they 1) encode the visual features into BEV space with instance- and position-aware subnets, 2) have an attention-based progressive refinement of hierarchical representations of increasingly-resolved map elements, and 3) a DETR-inspired loss formulation to associate predicted map elements to GT map elements, and then minimizing the associated polylines via a loss formulation over vertices and edges.

The results on nuScenes, especially for long-range detection [-60,60] , show a significant improvement to related work, while enjoying some speedups. The authors also provide a (rather short) ablation study about the impact of individual components, as well as the impact of varying vertex densities at the 6 progressive stages.

**Summary Of Recommendation:**

Overall I am inclined to accept the paper, as it's a sound and properly-executed idea. The paper writing itself has some flaws, though:
1) proper space utilization, where some explanations could have been shortened and others would have needed more details
2) relatively weak evaluation section, slightly connected with space issues stemming from 1).

I would suggest some reorganization of the writing, and a more detailed ablation, would make this paper a good addition to the conference.

---

> ### Author Response · Authors · 2023-08-13
> **Response to Reviewer 3QMy**
>
> **Q3.** The polyline loss seems interesting enough, but I see no clear benefit of the edge loss without any ablation study. Are there pathological cases where the vertex loss would fail to optimize towards the optimal configuration, and only the edge loss would provide enough signal to resolve a tie? Or is the idea to compensate for induced errors in GT resampling?
>
> **A3.** We greatly appreciate your astute observation. To address the merit of the edge loss, we've taken your suggestion and conducted ablation experiments, incorporating our designed edge loss for a more comprehensive analysis. Our edge loss is a composite of three crucial components: loss for newly added vertices relative to their corresponding edges, loss concerning edge slopes, and loss involving the angles formed by three consecutive vertices. The first component is particularly indispensable, as it directly contributes to the regression of map element shapes. We would like to illustrate the effect of the latter two losses on the model's performance.
>
> Initially, our perspective centered on supervising solely the vertices, presuming that it would suffice to guide shape restoration. However, this approach yielded weak constraints on the overall map element structure. The reliance on the distance between predicted and target points led to instances where originally straight lines devolved into zigzag shapes, misaligning with map element characteristics. To counteract this effect, we introduced the edge loss, effectively limiting the slope between adjacent points and the angle formed by three consecutive points. This augmentation significantly contributes to the faithful restoration of map element shapes.
>
> |  | with edge loss | w/o edge loss |
> | :-------:|:--------:|:--------:|
> | Ped Crossing | 44.8 | 43.9 |
> | Divider | 49.0 | 47.6|
> | Boundary | 43.1 | 42.7 |
> | mAP | 45.6 | 44.7|
>
> `We've meticulously documented the results of these experiments in Appendix B.2`, with the aim of offering a comprehensive view of our approach's efficacy. Your feedback serves as an invaluable catalyst for refining our methodology.
>
> We thank you again for your careful review. `Following your suggestion, we diligently conducted a thorough review of the methods section to identify any potential syntax errors. We also made adjustments to the overall layout of the paper to enhance its readability. Additionally, we have incorporated a visual representation of the progressive supervision as per your recommendation.` The revised version of the article has been uploaded and is available in the attached file. Please let us know if this addresses all of your questions and suggestions.

---

### Official Review · Reviewer_Z2cB · 2023-07-19

**Confidence:** 4
**Originality:** Good
**Technical Quality:** Good
**Clarity Of Presentation:** Good
**Impact:** 3

**Recommendation:**

Weak Reject: I recommend rejecting the paper, but will not argue for my recommendation if the majority of other reviewers have a different opinion.

**Review:**

The authors propose an intuitive and logical approach to online map learning and do well at explaining the proposed transformer-based approach. The authors are able to clearly point out what differentiates their paper from others.


Strengths:
- The authors present a promising approach and show how advances from transformer-based segmentation research can effectively be applied to the task of online map learning.
- I think the progressive supervision scheme is a very interpretable way of generating predictions and illustrates how real-time applications might benefit already from coarser representations.
- The ablation study seems to clearly show the effects of different design decisions regarding the hierarchical representation.


Weaknesses:

- I guess MapTR* was not fine-tuned for a different perception range. In contrast, I do not see any evidence that you did / did not use different hyperparameters for larger evaluation ranges.
- There is no comparison in terms of the number of parameters compared to MapTR. Since the successive decoding is your main contribution this needs a more thorough analysis. Since your results are rather incremental, how can we be sure its just not just because of a significantly larger modeling capability of the chosen network architecture in comparison to MapTR.


**Quality Of The Limitations Section:**

Additional details required

**Questions For Rebuttal:**

- Please provide some results on how your approach would work in an online map-preserving setting that retains + integrates previous predictions.
- Could you please delineate/portray/plot a sample prediction that suffers from severe occlusions? I would expect a collapsing prediction. How would you circumvent this in general? Do you have any kind of uncertainty estimate that could help here? Imagine a downstream task that uses map predictions. Should it rather utilize the current and potentially flawed prediction, or should it stick with a previous one + transformation given odometry is available?
- As this is a robotics-related conference, I would be interested in seeing some predictions under adverse weather conditions, if possible.
- Will you publish your code as part of this work? We usually expect that in conferences such as CoRL.


**Robotics Focus:**

Relevant but unlikely to deploy to hardware in near future

**Summary Of Paper:**

This paper presents an online approach to learning scalable map representations for HD map construction with a particular focus on large sensing distances. The authors motivate their approach by showing how especially axis-parallel objects that might span multiple distinct camera views are hard to capture using standard object detection formulations in comparison to vectorized approaches. First, the authors follow BEVFormer for encoding surround view imagery into BEV features that are both instance- and position-aware. Next, a set of queries is generated to predict coarse map elements. A so-called progressive supervision scheme is proposed in order to turn coarse maps into fine-grained vector-based map representations. By averaging out existing queries per step, new mean queries increase the number of vertices and thus increase the map resolution per step. The authors evaluate their architecture on the NuScenes validation-set and outperform MapTR on two distinct evaluation ranges.

**Summary Of Recommendation:**

The paper is well-motivated and shows interesting contributions in the field of online HD mapping. My main doubts are with respect to the network size and its comparison to MapTR as well as some proper analysis of failure cases from a robotics perspective. I am willing to raise my recommendation if my concerns are addressed during the rebuttal stage.

---

> ### Author Response · Authors · 2023-08-13
> **Response to Reviewer Z2cB**
>
> We would like to provide further clarification regarding your concern about the comparison with MapTR.
>
> **Q5.** I guess MapTR* was not fine-tuned for a different perception range. In contrast, I do not see any evidence that you did / did not use different hyperparameters for larger evaluation ranges.
>
> **A5.** Indeed, MapTR* is derived from MapTR by solely adjusting the perceived range, without specific modifications to hyperparameters. Similarly, our proposed ScalableMap is trained across various perceptual ranges with consistent hyperparameter settings. Our emphasis lies in leveraging the characteristics of map elements to enhance model convergence, rather than solely augmenting model expressiveness through parameter expansion. As a result, we achieve not only improved reasoning efficiency but also enhanced performance.
>
> **Q6.** There is no comparison in terms of the number of parameters compared to MapTR. Since the successive decoding is your main contribution, this needs a more thorough analysis. Since your results are rather incremental, how can we be sure it’s just not just because of a significantly larger modeling capability of the chosen network architecture in comparison to MapTR.
>
> **A6.** You've pointed out a crucial aspect that we unintentionally overlooked, and we appreciate your valuable input. To provide clarity, MapTR encompasses 36 million parameters, while ScalableMap comprises 48 million parameters. This increase is primarily attributed to the employment of a fully connected layer that fuses image characteristics from various perspectives. Importantly, our approach doesn't aim to enhance model expressiveness by disproportionately inflating specific parameter counts. Instead, we focus on the intrinsic structural characteristics of vectorized map elements, aiming to facilitate the model's learning process. In essence, our paper is rooted in the inherent attributes of vectorized map elements themselves.
>
> We are genuinely grateful for your detailed feedback, and your engagement is invaluable in guiding us towards the refinement and enrichment of our research.

---

### Official Review · Reviewer_f6RE · 2023-07-19

**Confidence:** 4
**Originality:** Good
**Technical Quality:** Very Good
**Clarity Of Presentation:** Very Good
**Impact:** 4

**Recommendation:**

Weak Accept: I recommend accepting the paper, but will not argue for my recommendation if the majority of other reviewers have a different opinion.

**Review:**

- Originality: the paper is well motivated, carving out the novel missing piece in the related literature, which is a structured prediction approach targeting long range map construction. Related work is widely surveyed and well cited.
- Quality: the proposed framework is technically sound, backed up by comprehensive evaluations and ablation study.
- Clarity: the paper is well written, easy to follow.
- Significance: the work is addressing an important yet crowded problem area. The significance is incremental but important and difficult.
- Relevance: the online mapping problem is critical in the automotive industry. The paper is definitely on the right path.
- Limitations: Discussion section clearly states the limitation of camera-only, occlusion, being calibration dependent,not using prior knowledge, etc.


**Quality Of The Limitations Section:**

Limitations are addressed clearly

**Questions For Rebuttal:**

- It is a little bit unclear what is the semantic segmentation head doing, how is it fusing position-aware and instance-aware embeddings. Are they aligned by semantics? It would be great if that particular paragraph can be expanded a bit.
- Minor missing reference in section 4.1:  “Following previous works  [12?,15]”


**Robotics Focus:**

Highly relevant to robotics but no hardware experiments

**Summary Of Paper:**

The paper proposes ScalableMap, an online vectorized HD map prediction approach that leverages structure guidance and progressive decoding and supervision, targeting enhancing long range prediction quality. The approach extract position-aware features using BEVFormer and instance-aware features using MLPs, fusing them with semantic segmentation head, followed by a cascade of transformers for different resolution decoding and supervision. Evaluation on nuScens dataset shows that the approach beats state-of-the-art by 6.5mAP while maintaining a lightweight footprint, executing at 18FPS.


**Summary Of Recommendation:**

The paper is addressing an important problem of online long range mapping for autonomous vehicles. The proposed approach is evaluated better than existing state-of-the-art while maintaining a light footprint and high FPS.

---

### Decision · Program_Chairs · 2023-08-30

**Decision:**

Accept (Poster)

**Comment:**

This paper proposes a vectorized HD map prediction approach focused on improving the long-range prediction performance. The reviewers find the paper well written and well motivated, addressing an important problem. There were several concerns raised by the reviewers during the initial review, most of which were addressed in the rebuttal. A few minor concerns exist, e.g, didn't add model size comparison to the paper, although mentioned in the rebuttal.